# Hidden risks associated with occupational pesticide exposure in women with breast cancer: High frequency of the Luminal B molecular subtype and occurrence of poor prognostic features

Isabella C. Cazagranda[1], Rafaela Frederico de Almeida[2], Lucca L. Smaniotto[2], Maria Paula de Andrade Berny[2], Carolina Coradi[2], Daniel Rech[2], Carolina Panis[2]*, Guilherme F. Silveira [1]*

1 Grupo de Imunologia Molecular, Celular e Inteligência Artificial, Instituto Carlos Chagas, Fundação Oswaldo Cruz (FIOCRUZ-PR), Curitiba, Brazil, 2 Laboratório de Biologia de Tumores, Centro de Ciências da Saúde, Universidade Estadual do Oeste do Paraná, Campus de Francisco Beltrão, Francisco Beltrão, Paraná, Brazil

* carolpanis@hotmail.com (CP); guilherme.silveira@fiocruz.br (GFS)

## Abstract

Human pesticide exposure is a common event in countries with strength conventional agriculture, such as Brazil. Despite evidence on the negative impact of pesticides on human health, the country stands out among the top three pesticide consumers globally. The implications of this scenario on rural workers health, particularly women, is completely neglected, resulting in chronic illness such as breast cancer. Objective: In this study, we analyzed the impact of occupational/household chronic exposure to pesticides on the clinicopathological profile of breast cancer in rural women from Paraná southwest, a predominantly rural landscape with large pesticide uses. Methods: A total of 349 women were included in the study. After a structured interview, women were categorized as exposed (n = 208) or unexposed (n = 141) to pesticides. Clinicopathological data were collected from medical records. Descriptive and inferential statistical methods were used to characterize and compare the sample. The Chi-square test and Fisher's exact test were used to evaluate differences between the molecular subtypes and clinicopathological variables of patients. Results: Exposed patients had a prevalence of the Luminal B subtype (32.83%), while unexposed patients had a prevalence of the Luminal A molecular subtype (37.78%, p <= 0.05). Exposed patients also had higher disease recurrence (10.19%), chemoresistance (21.26%), than unexposed patients (p <= 0.05). Breast cancer patients exposed to pesticides were also more likely to have distant metastases (1.4 times) and lymph node invasion (1.3 times) compared to patients not exposed. Conclusions: These findings indicate that pesticide exposure favors the occurrence of more aggressive breast cancer.

**Data availability statement:** The original contributions presented in the study are included in the article Materials and Methods. Documentation and codes used in the analysis, are available at https://zenodo.org/records/12667841.

**Funding:** The author(s) declare that financial support was received for the research, authorship, and/or publication of this article. Part of this research was supported by Carlos Chagas Institute, Fiocruz/PR, Fundação Araucária (grant number 639/2022) and by Conselho Nacional de Desenvolvimento Científico e Tecnológico – CNPq (grants number 402364/2021-0, 441017/2023-1, and 305335/2021-9).

**Competing interests:** The authors declare that the research was conducted in the absence of any commercial or financial relationships that could be construed as a potential conflict of interest.

## Introduction

Pesticide use in agriculture began with the "Green Revolution" movement in the 50's, under the argument of increasing food production [1]. Unfortunately, at that time, there was no understanding of the potential risks of its use for the environment and human health [2].

Since then, several studies have demonstrated the risks attributable to pesticide exposure in human health, such as their carcinogenic potential, [3,4] endocrine disrupting properties [5], genotoxicity [6–8], and immunotoxicity [9–12], which can result in cancer. Several pesticides have been classified by the International Agency for Cancer Research (IARC) as potentially, probably, or proven carcinogens. A variety of cancers have been linked to pesticide exposure, including thyroid [13], colorectal [14], bladder [15], blood [16], brain [17], and breast cancer [18].

Pesticides are known endocrine disruptors and can influence the development of tumors in the female reproductive system [19], increase aromatase activity and estrogen production [20,21], reduce fertility [20], augment estrogen production [22], increase androgen availability [23,24], competitively bind to estrogen cell receptors [25], enhance proliferation of estrogen-sensitive cells, and inhibit corticosterone synthesis in the adrenal cortex [20,23,26]. Pesticides can elevate the risk of breast cancer through various mutagenic and nonmutagenic mechanisms, acting either directly as carcinogens or indirectly as biochemical modifiers and hormonal disruptors. The underlying mechanisms include endocrine disruption [27], genotoxicity [28], epigenetic alterations [29], enhanced cell migration, invasion, and stemness [30], angiogenesis [31], and tumor growth [32], among others. The Southeastern region of northern Paraná has a prevalence of glyphosate and atrazine use.

Glyphosate is an endocrine disrupting substance, a pivotal mechanism proposed to breast carcinogenesis. In normal mammary cells, it promotes estrogenic activity by mimicking 17β-estradiol (E2), activating estrogen receptor α (ERα), leading to phosphorylation, degradation, and increased transcriptional activity, as well as stimulating cell proliferation and growth [33], even at very low concentrations [34]. In breast cancer cells, glyphosate exposure leads to altered expression of cell proliferation-related genes [35], and dysregulation of key genes involved in tumor aggressiveness and metastasis, such as HIF-1 [36].

Atrazine, also exhibit endocrine-disrupting effects, particularly on mammary gland development in murines [37], delaying mammary gland development and increasing steroidogenesis and hormone levels [38]. Hormone deregulation is also documented for glyphosate and atrazine human exposures [39]. In breast cancer cells, atrazine alters protein expression and modulates antioxidant defense gene expression, promoting genomic instability and oxidative stress-induced damage [40], a recognized mechanism for breast cancer development and progression, also linked to immune deregulation in patients [41–43] and inflammatory changes in normal mammary tissue in exposed women [44].

2,4-dichlorophenoxyacetic (2,4-D) is the second most widely traded pesticide in Brazil [45]. The IARC classifies it as possibly carcinogenic [46] and, in humans, exposure has been associated with an increased risk of mesothelioma and

non-Hodgkin lymphoma [47,48]. Additionally, women occupationally exposed to areas where glyphosate, atrazine, and 2,4-D are predominantly used have a higher risk of developing breast cancer [49].

In the context of pesticide health risks, it is important to note the growth of the feminization movement in agriculture; it has been estimated that women represent 43% of the world's agricultural workforce [50]. This trend has been observed in several regions of the world, such as in the European Union, where women represent 29% of rural workers [51], Brazil, where they represent 45%, and certain regions of Africa and Asia, where women's representation can reach up to 60% [52]. The feminization of agriculture may lead to an increase in the incidence of cancer in women.

Due to the extensive exposure of rural women in agriculture and the endocrine disruption properties of pesticides, attention has been drawn to the incidence of female tumors, such as breast cancer. Several studies have addressed the relationship between pesticide exposure and the increased risk of developing this pathology [53–55]. Endocrine deregulation can occur by mainly two exposure models: programming, which modifies tissues during embryonic development until puberty, making them susceptible to cancer, and worsening, where subsequent exposure leads to malignant evolution of precancerous cells or benign lesions [19].

Considering that little is known about the impact of chronic and continued occupational pesticide exposure population from Southwest region of Paraná, Brazil, on the clinicopathological profiling of breast cancer, the present study focused on understanding this issue. To reach this goal, we performed extensive data collection from patients diagnosed with breast cancer, occupationally exposed or not to pesticides. Information concerning patient profiles and tumor characteristics was obtained and analyzed using descriptive and inferential statistical methods, with the aim of determining a clinicopathological signature associated with pesticide exposure.

## Methods

### Study design and data collection

This is a descriptive, cross-sectional, and quantitative exploratory study with the objective of determining a clinicopathological signature associated with pesticide exposure in rural women diagnosed with breast cancer. Fig 1 illustrates the study design.

This study complied with the national and international regulatory standards for research involving humans and was approved by the Research Ethics Committee (CEP) of the State University of Western Paraná (UNIOESTE), under the number CAAE 35524814.4.0000.0107. The study was conducted in accordance with the local legislation and institutional requirements. The participants provided their written informed consent to participate in this study. All volunteers signed a free and informed consent form, and the study did not include minors. The patients were recruited for this study and the data was accessed for research purposes from 05/27/2015 to 04/20/2023. Documentation and codes used in the analysis, are available at https://zenodo.org/records/12667841, this information could not identify individual participants, following the rules of the Ethics Committee.

A total of 923 women attended at the Francisco Beltrão Cancer Hospital (Ceonc), a public Hospital Cancer Center that attends patients from 27 municipalities of Parana state southwest, from May 2015 to April 2023 with images suggestive of breast lesions identified by mammograms and ultrasound were included. The study comprised the population belonging to the Eighth Paraná Health Region, which includes 27 municipalities (Fig 1) characterized predominantly by rural family work in the Eighth Regional Health Region of Paraná, comprises about 500.000 inhabitants. We chose this region to develop the study because Paraná is the state that sells the fourth highest amount of pesticides in Brazil [56]. This reflects an extensive use of pesticides in the state's agricultural activities, which play a significant role in the Gross Domestic Product (GDP) of the 27 municipalities that make up this health area [56]. More than 50% of the region's inhabitants engage in agricultural activities, with a particular focus on family farming. This population is subject to considerable pesticide exposure, especially glyphosate, atrazine, and 2,4-dichlorophenoxyacetic (2,4-D), which are widely used in soybean, corn, and wheat monocultures in the region [56]. Fig 1A depicts the correlation between breast cancer cases and the amount of pesticides used by municipalities in the Eighth Health Region of Paraná.

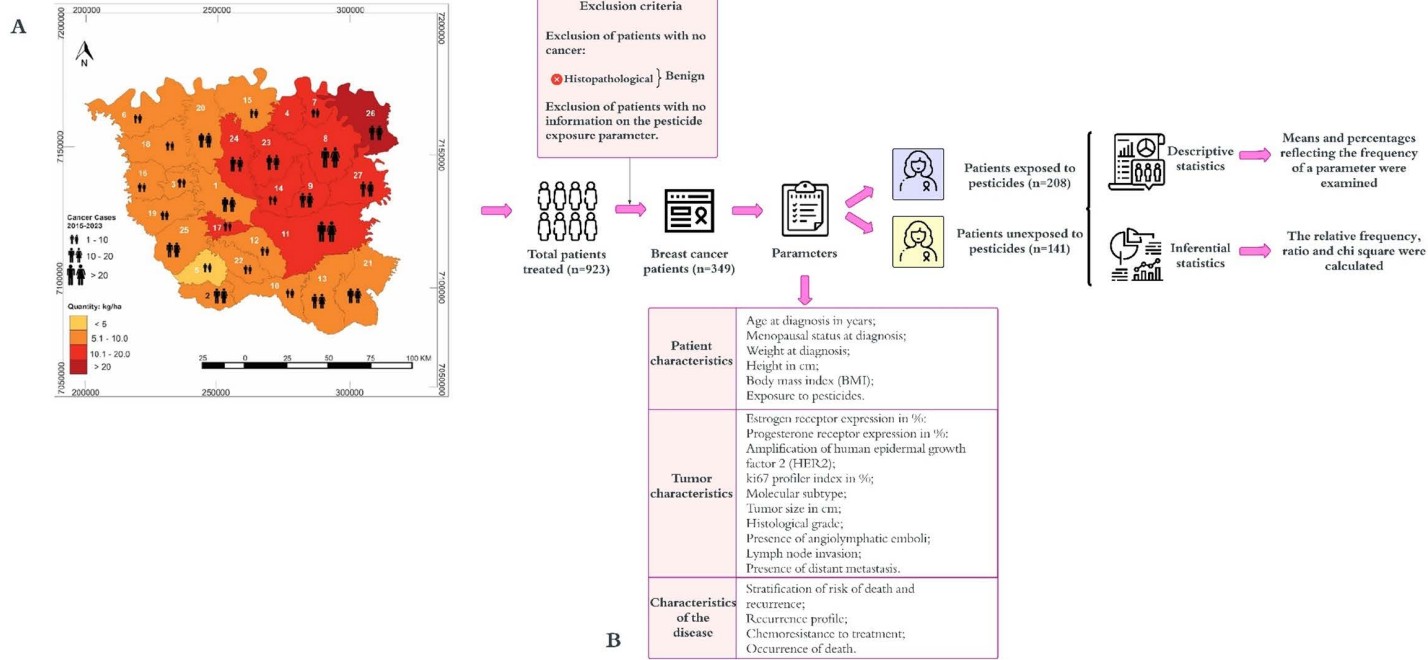

**Fig 1. Study design.** Of the 923 patients with images suggestive of breast lesions identified by mammograms and ultrasound, 349 patients were included in the study analyses because they had a diagnosis of breast cancer determined by a pathologist. To achieve the study's objective, descriptive and inferential statistical analyses were performed on the segregated population of patients exposed and unexposed to pesticides. The municipalities present in the work was 1 – Ampére, 2 – Barracão, 3 – Bela Vista da Caroba, 4 – Boa Esperança do Iguaçu, 5 – Bom Jesus do Sul, 6 – Capanema, 7 – Cruzeiro do Iguaçu, 8 – Dois Vizinhos, 9 – Éneas Marques, 10 – Flor da Serra Do Sul, 11 – Francisco Beltrão, 12 – Manfrinópolis, 13 – Marmeleiro, 14 – Nova Esperança do Sudoeste, 15 – Nova Prata do Iguaçu, 16 – Pérola D'Oeste, 17 – Pinhal de São Bento, 18 – Planalto; 19 – Pranchita, 20 – Realeza, 21 – Renascença, 22 – Salgado Filho, 23 – Salto do Lontra, 24 – Santa Izabel do Oeste, 25 – Santo Antônio do Sudoeste, 26 – São Jorge D'Oeste, and 27 – Verê. A table with the profile and grouping of the parameters analyzed in the study. The characteristics of the patients include the following parameters: age at diagnosis in years, menopausal status at diagnosis; weight at diagnosis, height in cm; body mass index (BMI); exposure to pesticides. Among the tumor characteristics are the following parameters: estrogen receptor expression in %; progesterone receptor expression in %; amplification of human epidermal growth factor 2 (HER2); Ki67 profiler index in %; molecular subtype; tumor size in mm; histological grade; presence of angiolymphatic emboli; lymph node invasion; presence of distant metastasis. Among the characteristics of the disease are the following parameters: stratification of the risk of death and recurrence, recurrence profile, chemoresistance to treatment and occurrence of death.

For breast cancer diagnosis, a biopsy of the suspicious lesion was analyzed by a pathologist, followed by anatomo-pathological analysis and immunohistochemistry. After excluding patients with benign lesions, 386 patients having a breast cancer diagnosis were included in the study. In an initial check of the database of 386 patients, we noted inconsistencies in the data (some of the variables did not contain complete values, or the values were out of scale, indicating collection error), so 37 (9,5%) of the patients were removed, for a total of 349.

**Pesticide exposure assessment.** To characterize exposure, we previously performed a 2-year study aiming to get detailed information about patients' exposure profile. To reach this goal, patients were invited to complete a comprehensive questionnaire with 61 questions covering their current and past occupational history. Based on their responses, we categorized the study population as either occupationally exposed or unexposed to pesticides.

Women in the exposed group reported spending at least 50% of their lives working with pesticides and having direct contact with these substances at least once a week. Their exposure involved washing clothes and personal protective equipment (PPE) contaminated with pesticides worn by family members who applied them; preparing and diluting concentrated pesticides, and/or assisting in the spraying of diluted pesticides on crops, typically for 4–8 hours per day over 2–3 consecutive days, every 1–2 weeks.

Given that Brazil's arable land is nearly fully cultivated and that the study area is a key agricultural region in Paraná State (which ranks second in the state's GDP), occupational exposure among these patients is notably intense. Furthermore, 94% of the women in the exposed group reported performing these tasks without using PPE, not even gloves. As pesticides are primarily absorbed through the skin, this chronic and prolonged exposure represents a significant contamination route, surpassing potential exposure from food or water sources.

In contrast, women in the unexposed group did not engage in any rural work involving pesticides and reported no current or past occupational contact with these substances, despite being from the same region as the exposed patients. They had no history of washing pesticide-contaminated clothes or PPE and never participated in pesticide application on crops. These distinct occupational histories qualified them for the unexposed group.

It is important to note that food consumption habits were similar in both groups, and both were supplied by the same hydrographic water system—the left strand of the Lower Course of the Iguaçu River Basin. Even if this water system were contaminated with pesticides, the occupational exposure in the exposed group would still far outweigh this potential source due to its intensity and severity.

Given the chronic and intense nature of occupational pesticide exposure, we are confident that our study provides substantial evidence of distinct exposure conditions between the two groups. Despite similar geographic locations, genetic backgrounds, and cultural and dietary habits, the unexposed group had no occupational pesticide exposure. This distinction emphasizes our focus on the impact of occupational exposure rather than environmental contamination from food or water.

The following data were used as defining parameters of patient characteristics: age in years at diagnosis and menopausal (dichotomized as presence and absence) status at diagnosis. The patient's weight at diagnosis was also obtained in kilograms (kg), height in meters (m), and body mass index (BMI) in kg/m$^2$. Data were grouped into three categories: patient characteristics, tumor characteristics, and disease characteristics (Fig 1B).

Data concerning the occupational pesticide exposure was obtained by using a standardized data collection instrument validated for this purpose [57]. The exposure criteria was based on continuous, unprotected, and direct handling of pesticides. The exposed group consisted of patients with a history of direct handling of pesticides without use of protective gloves during preparation and/or dilution of pesticide solution, application of pesticides, and/or decontamination of personal protective equipment and/or washing of clothes used during spraying, who reported living at least 50% of their lives under direct handling of such substances at least twice a week for every week of the year. The unexposed group is comprised of urban workers with no previous or current history of occupational pesticide exposure. Based on this, patients were categorized as occupationally exposed (n = 208) or unexposed to pesticides (n = 141), all Caucasians.

**Tumor characteristics.** The following parameters were considered as tumor characteristics: estrogen (ER) and progesterone (PR) receptors' expression (%); amplification of the epidermal human growth factor receptor 2 (HER2); proliferation index Ki67 in %; molecular subtyping of breast tumors considering the classes Luminal A = any positivity for ER and/or PR and ki67 below/equal to 14%, Luminal B = any positivity for ER and/or PR and ki67 above 14%, HER2-amplified = ER/PR negative, any ki67 value and presence of amplification for HER2, and Triple Negative = ER/PR/HER2 negative and any ki67 value (as described by the St. Gallen Consensus) [58]; tumor size represented in mm, histological grade categorized as low (grades 1 and 2) and high (grade 3). Lymph node invasion, presence of angiolymphatic emboli, and occurrence of distant metastases were dichotomized as presence or absence.

**Disease characteristics.** The parameters considered as disease characteristics were: risk stratification for death and recurrence (low, intermediate, and high risk, as described in Joint Ordinance No. 5 of April 18, 2019) [59], chemoresistance development (based on the RECIST 1.1 guideline [60] and [61]), disease recurrence, and death. All data were dichotomized as presence or absence. In the present dataset we do not have information regarding education level/ subject income/ ancestry, which may be a limitation of this work.

Patients with early-stage breast cancer who underwent neoadjuvant chemotherapy and were subsequently followed for disease recurrence were included in the study. Individuals who developed recurrence or systemic progression received

adjuvant therapy. Chemotherapy data were retrieved from the Authorization for Outpatient Procedures (APAC) system. The chemotherapy protocols administered to the patients included three main regimens: AC-T, CMF, and TCH. The AC-T protocol consisted of doxorubicin (60 mg/m$^2$, day 1) combined with cyclophosphamide (600 mg/m$^2$, day 1) every 21 days for four cycles, followed by paclitaxel (175 mg/m$^2$) administered either weekly for 12 weeks or every 21 days after completion of the AC cycles. The CMF regimen comprised cyclophosphamide (600 mg/m$^2$, day 1), methotrexate (40 mg/m$^2$, day 1), and 5-fluorouracil (600 mg/m$^2$, day 1), repeated every 21 days for six cycles. Patients receiving the TCH regimen were treated with trastuzumab (loading dose of 8 mg/kg, followed by 6 mg/kg on day 1 every 21 days for 12 months), carboplatin (AUC 6, day 1 every 21 days for six cycles), and docetaxel (75 mg/m$^2$, day 1).

For patient classification, individuals who did not achieve a response to neoadjuvant chemotherapy were defined as chemoresistant. The assessment followed the RECIST (Response Evaluation Criteria in Solid Tumours, available at: https://recist.eortc.org/recist-1-1-2/) guidelines.

Imaging reports from mammography, diagnostic ultrasound, breast MRI, CT, and PET-CT during follow-up were analyzed by clinicians in accordance with RECIST recommendations to establish baseline and post-treatment status. For each patient, the same imaging modality was applied at baseline and follow-up to ensure consistency.

Treatment response was categorized as follows: complete response, defined as the disappearance of all target lesions; partial response, defined as at least a 30% reduction in target lesion size without evidence of new lesions; progressive disease, defined as a minimum 20% increase in lesion size relative to baseline and/or the appearance of new lesions in the breast or distant organs; and stable disease, defined as insufficient shrinkage to meet partial response criteria and insufficient increase to meet progression criteria. Patients were monitored over a five-year follow-up period. Based on these criteria, patients were classified as responsive (complete response) or chemoresistant (partial response, progressive disease, or stable disease).

Concerning the risk stratification for recurrence and death [59], patients were stratified into risk categories based on lymph node status, tumor size, histological grade, hormone receptor expression, HER-2 status, molecular subtype, and age. Low-risk patients had negative lymph nodes and met all of the following criteria: tumor size (pT) under 2 cm, histological grade 1, estrogen receptor (ER) or progesterone receptor (PR) positive, HER-2 negative, molecular subtype luminal A, and age ≥ 35 years. Intermediate-risk patients had negative lymph nodes but met at least one of the following criteria: tumor size > 2 cm, histological grade 2–3, ER or PR negative, molecular subtype luminal B (HER-2 negative), age < 35 years, or 1–3 positive lymph nodes if ER and PR were positive. High-risk patients included those with 4 or more positive lymph nodes, lymph node-negative tumors that were triple-negative (ER, PR, and HER-2 negative) with pT > 2 cm, or lymph node-negative tumors with pT > 1 cm and HER-2 positive status.

## Data analysis

A descriptive statistical analysis of the data was performed, including characteristics of the disease, tumor and patient, where the frequency of each of the parameters analyzed was calculated (Table 1). In the inferential statistics, the Chi-square test and Fisher's exact test were performed – for samples with n < 5 (Table 2). A p < 0.05 was considered significant. A ratio was also calculated using the absolute frequency of each molecular subtype (Luminal A, Luminal B, HER2-amplified and Triple-negative) for each of the clinicopathological parameters (Table 2). For this, the absolute frequency value of one molecular subtype was divided by the other. The parameters that presented significance were subjected to the Chi-square contingency test to confirm their independence. All analyses were processed in Python version 3.10.12.

## Study limitations

This work has some limitations. In previous classifications, the Luminal B subtype of breast cancer was often defined as estrogen receptor (ER)-positive and HER2-amplified, meaning that tumors expressing concomitant hormone receptors and HER2 overexpression were categorized as Luminal B (HER2-positive). Currently, the Luminal B subtype is identified

**Table 1. Frequency of clinicopathological characteristics of patients included in the study. Some of the patients present in the database did not present all the information on all the variables.**

| | Total population (n = 349) | Population exposed (n = 208) | Population unexposed (n = 141) |
|---|---|---|---|
| Patient Characteristics | | | |
| Mean age at diagnosis (years) | 56 (n = 331) | 56 (n = 197) | 56 (n = 134) |
| Menopausal status (at diagnosis) | 67.75% (n = 219) | 67.71% (n = 130) | 67.42% (n = 89) |
| BMI (kg/m²) | 27.95 (n = 247) | 27.82 (n = 150) | 27.92 (n = 97) |
| Occupational pesticide exposure | 59.60% (n = 208) | 100.0% (n = 208) | 0.0% (n = 0) |
| Tumor Characteristics | | | |
| Estrogen receptor expression | 72.10% (n = 261) | 67.49% (n = 137) | 78.10% (n = 107) |
| Progesterone receptor expression | 50.83% (n = 183) | 45.05% (n = 91) | 57.35% (n = 78) |
| HER2 | 13.41% (n = 48) | 12.50% (n = 25) | 13.97% (n = 19) |
| KI-67 | 62.67% (n = 225) | 62.67% (n = 225) | 62.67% (n = 225) |
| Average tumor size (mm) | 29.21 (n = 357) | 30.82 (n = 201) | 28.11 (n = 134) |
| Molecular subtyping | | | |
| Luminal A | 33.24% (n = 112) | 30.81% (n = 61) | 37.78% (n = 51) |
| Luminal B | 34.37% (n = 113) | 33.33% (n = 66) | 34.81% (n = 47) |
| HER2-amplified | 16.62% (n = 54) | 16.67% (n = 33) | 15.56% (n = 21) |
| Triple-negative | 15.77% (n = 54) | 19.19% (n = 38) | 11.85% (n = 16) |
| Grade | | | |
| Grade 1 | 28.21% (n = 101) | 29.15% (n = 58) | 29.20% (n = 40) |
| Grade 2 | 51.68% (n = 185) | 52.26% (n = 104) | 48.91% (n = 67) |
| Grade 3 | 20.11% (n = 72) | 18.59% (n = 37) | 21.90% (n = 30) |
| Presence of angiolymphatic emboli | 25.52% (n = 86) | 27.51% (n = 52) | 24.06% (n = 32) |
| Lymph node positivity | 36.86% (n = 108) | 40.35% (n = 69) | 32.69% (n = 34) |
| Distant metastasis | 41.13% (n = 116) | 46.39% (n = 77) | 35.24% (n = 37) |
| Disease characteristics | | | |
| Stratification of death risk and recurrence | | | |
| Low risk | 8.25% (n = 25) | 8.24% (n = 15) | 8.26% (n = 10) |
| Medium risk | 55.87% (n = 168) | 52.20% (n = 95) | 60.33% (n = 73) |
| High risk | 35.87% (n = 100) | 39.56% (n = 72) | 31.40% (n = 38) |
| Recurrence | 9.36% (n = 374) | 10.19% (n = 206) | 9.93% (n = 141) |
| Chemoresistance | 18.97% (n = 70) | 21.26% (n = 44) | 17.27% (n = 24) |

The parameters age at diagnosis (years), BMI (kg/m²), and tumor size are presented by the population mean analyzed, and the parameters occupational pesticide exposure, molecular subtyping, histological grade, stratification of death risk and recurrence, recurrence, and chemoresistance are presented as a percentage of the analyzed group in relation to the total number of individuals in the population. The number of patients in each variable is different due to missing values (all variables have less than 15% of missing values, except for the variable BMI with 28%). n represents the number of patients with the positive clinicopathological parameter evaluated.

independently of HER2 status, being characterized instead by positive hormone receptor expression (ER and/or PR) combined with a high proliferative index (Ki-67 > 14%), regardless of HER2 amplification. The initial design of our study did not aim to include patients with the Luminal-HER2 subtype due to the small number of cases in this subgroup; therefore, this subgroup was excluded from the final analysis to avoid biased interpretations based on an insufficient sample size. Since their data were not collected, they could not be included in the present study. The results were presented considering only the four main breast cancer subtypes, considering the St. Gallen International Expert Consensus [62] (Table 3), which recognizes that molecular classification based on immunohistochemistry groups breast tumors into these four main

**Table 2. Association between molecular subtypes and clinical pathological variables with significance in breast cancer patients occupationally exposed and not exposed to pesticides.**

| Parameter | Occupational pesticide exposure | Lum B | Lum A | Ratio Lum B/ Lum A | p-value | Lum B | HER2-amplified | Ratio Lum B/ HER2-amplified | p-value | Lum B | Triple-negative | Ratio Lum B/ Triple-negative | p-value |
|---|---|---|---|---|---|---|---|---|---|---|---|---|---|
| Presence of angiolymphatic emboli | | | | | | | | | | | | | |
| | Exposed | 18 | 15 | 1.2 | 0.6 | 18 | 6 | 3 | 0.01 | 18 | 13 | 1.38 | 0.37 |
| | Unexposed | 16 | 7 | 2.29 | 0.06 | 16 | 4 | 4 | 0.26 α | 16 | 4 | 4 | 0.75 α |
| Lymph node invasion | | | | | | | | | | | | | |
| | Exposed | 30 | 19 | 1.58 | 0.12 | 30 | 9 | 3.33 | <0.01 | 30 | 10 | 3 | <0.01 |
| | Unexposed | 15 | 9 | 1.67 | 0.22 | 15 | 3 | 5 | 0.73 α | 15 | 5 | 3 | 0.03 |
| Distant metastasis | | | | | | | | | | | | | |
| | Exposed | 30 | 22 | 1.36 | 0.27 | 30 | 13 | 2.31 | <0.01 | 30 | 11 | 2.73 | <0.01 |
| | Unexposed | 16 | 10 | 1.6 | 0.24 | 16 | 4 | 4 | 0.53 α | 16 | 5 | 3.2 | 0.02 |
| Menopausal status | | | | | | | | | | | | | |
| | Exposed | 46 | 42 | 1.1 | 0.67 | 46 | 14 | 3.29 | <0.01 | 46 | 25 | 1.84 | 0.01 |
| | Unexposed | 25 | 39 | 0.64 | 0.08 | 25 | 9 | 2.78 | <0.01 | 25 | 12 | 2.08 | 0.03 |

The columns with the molecular subtypes (Luminal A, Luminal B, HER2-amplified and Triple-negative) present the absolute frequency values for each of the clinicopathological parameters. The ratio was calculated by dividing the absolute frequency of the 1st molecular subtype by the absolute frequency of the 2nd molecular subtype of the column title. The Chi-square test and Fisher's exact test were calculated using the comparison of the 1st and 2nd molecular subtypes reported in the ratio column arranged to the left of the p-value column. A $p < 0.05$ was considered significant. The abbreviation Lum corresponds to the Luminal molecular subtype. α represents that the p-values were calculated using Fisher's exact test, the other values were calculated using the chi-square test for independence.

**Table 3. Definition of subtypes of breast cancer included in the study accordingly to the St. Gallen classification (Goldhirsch et al., 2013).**

| Subtypes of breast cancer | ER and PR | HER2 | Ki67 |
|---|---|---|---|
| Luminal A | ER + and/or PR + | Negative | Ki-67 < 14% |
| Luminal B | ER+ and/or PR+ | Negative | Ki-67 ≥ 14% |
| HER2 enriched | ER-, PR- | Positive | Any Ki-67 |
| Triple negative | ER-, PR- | Negative | Any Ki-67 |

Legend: Negative (-), positive (+). Luminal-HER2 patients (ER+, PR+, and HER2+) were not included in the study.

categories. We believe this decision does not compromise the methodological rigor or the scientific merit of our study [63,64] (Coates 2015, Burstein 2023).

The question regarding the follow-up, was used only the RECIST (Response Evaluation Criteria In Solid Tumors) criteria for the follow-up period. This wis a globally recognized standard for evaluating a patient's tumor response to treatment, used to assess whether tumors shrank, stayed the same, or grew. It was the standard used by our medical team for years, which made it impractical to apply another criterion retroactively. The low mortality rate observed in our study was because all patients were in Stage II. Stage II cancer was considered an early stage and was often curable

with treatments like surgery, radiation, or chemotherapy, which was a reason for the lower mortality rate. Our study only considered deaths from cancer, and we did not have access to information on deaths from other causes.

## Results

### Breast cancer with prevalence of the Luminal B molecular subtype was observed in occupationally exposed to pesticides patients

Table 1 shows the clinicopathological data of the study population according to their pesticide exposure profile. Breast cancer patients (n = 349) had a mean age of 56 years, ranging from 22 to 96 years. Average BMI was 27.95 kg/m$^2$ (16.4–51.26 kg/m$^2$). About 60% of the patients were occupationally exposed to pesticides. Of this population, 8.25% were stratified as low risk for death and recurrence, 55.87% into intermediate risk, and 35.87% were classified as high risk. Regarding the molecular subtype, 33.24% of the patients were classified as Luminal A, 34.37% as Luminal B, 16.62% as HER2-amplified, and 15.77% as Triple-Negative. Also, 28.21% of the tumors were grade 1, 51.68% were grade 2, and 20.11% were grade 3. About 7% of the patients died, 9.36% of the patients had disease recurrence, and 18.97% of the patients developed chemoresistance.

The study population was divided based on occupational pesticide exposure profiles and showed summary measures similar to those of the total cohort (n = 349). We chose to perform an intragroup analysis initially to maintain the consistency of the patients' conditions. Among breast cancer patients occupationally exposed to pesticides (n = 208), the Luminal B molecular subtype was prevalent (33.33%), while unexposed patients (n = 141) predominantly exhibited the Luminal A subtype (37.78%). By comparing the prevalence of the Luminal A subtype between unexposed and exposed groups, it was found that unexposed patients were 1.2 times more likely to present with a molecular subtype associated with a better prognosis (calculated by dividing the percentage of Luminal A in unexposed patients (37.78%) by that in exposed patients (30.81%)). Additionally, the analysis of the Triple-negative subtype showed that patients exposed to pesticides were 1.5 times more likely to develop this subtype, which is associated with a worse prognosis, compared to unexposed patients (19.19% in the exposed group vs. 11.85% in the unexposed group). Furthermore, exposed patients demonstrated higher rates of recurrence (10.19%), chemoresistance (21.26%), and mortality (7.21%) compared to both unexposed patients and the total population, further indicating a worse prognosis.

### Patients exposed to pesticides are more likely to have distant and lymph nodal metastases

We sought to determine which characteristics were significantly distinct in breast cancer patients according to their occupational pesticide exposure profile. The statistically significant and clinically relevant results are summarized in Table 2. Other statistical values analyzed are present in the supplementary materials (S1 and S2 Tables in S1 File). All variables were evaluated using the Chi-square analysis or Fisher's test for comparisons among the molecular subtypes.

Unexposed patients at menopause showed an increased frequency of Luminal B compared to Triple-negative (p = 0.03 and ratio = 2.08) and HER2-amplified tumors (p=<0.01 and ratio = 2.78) (Table 2). Exposed patients at menopause showed increased Luminal B frequency compared to Triple-negative (p = 0.01 and ratio = 1.84) and HER2-amplified tumors (p=<0.01 and ratio = 3.29) (Table 2).

Breast cancer patients exposed to pesticides were 1.4 times more likely to develop metastasis compared unexposed patients. (Fig 2). Increased frequency of Luminal B tumors was observed in relation to HER2-amplified (p=<0.01 and ratio = 2.31) and Triple-negative (p=<0.01 and ratio = 2.73) in the exposed group (Table 2) regarding distant metastasis. For unexposed patients, there increased frequency of Luminal B tumors was found compared to Triple-negative (p = 0.02 and ratio = 3.20) (Table 2) when considering distant metastasis.

Regarding lymph nodal invasion, breast cancer patients exposed to pesticides were 1.3 times more likely of having a positive lymph node compared to unexposed patients (Fig 2) Increased frequency of Luminal B tumors compared

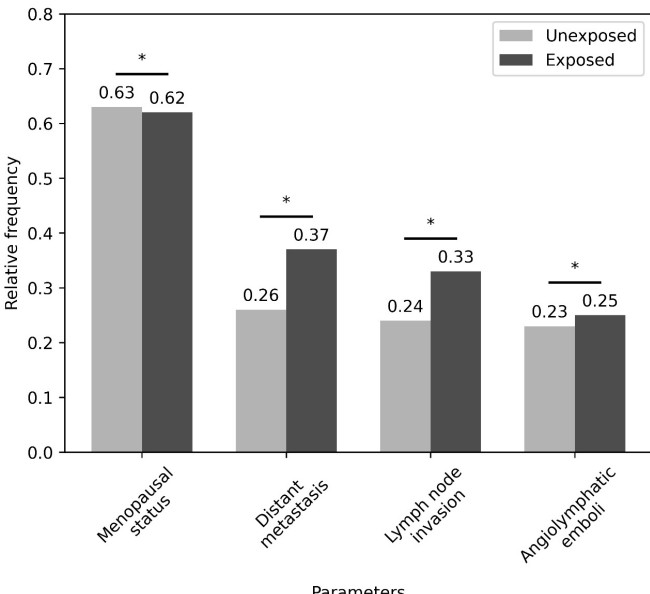

**Fig 2. Comparison of the relative frequency of parameters that showed significance for patients exposed and unexposed to pesticides.**
*represents that there is no statistical evidence of dependence between the variables using the chi-square test of independence.

to HER2-amplified molecular subtypes (p=<0.01 and ratio = 3.33) (Table 2) was identified. For unexposed patients, an increased frequency of Luminal B tumors compared to Triple-negative (p = 0.03 and ratio = 3.0) was identified (Table 2).

Concerning the presence of angiolymphatic emboli, unexposed patients did not show any significant differences in relation to the molecular subtype (Table 2), while exposed patients presented an increase in the frequency of Luminal B compared to HER2-amplified tumors (p = 0.01 and ratio = 3.0) (Table 2).

The parameters menopausal status, distant metastases, lymph node invasion and angiolymphatic emboli were subjected to the contingency chi-square test and confirmed their independence.

## Discussion

Our study indicates that occupational pesticide exposure is linked to the occurrence of breast tumors with more aggressive clinicopathological characteristics. In the exposed population, we observed an increased frequency of disease recurrence, chemoresistance, death, and predominance of the molecular subtype Luminal B, in intragroup analysis. To our knowledge, this is the first study that uses the described methodology to predict the relationship of variables related to breast cancer severity in a population categorized according to their pesticide exposure profile.

We identified that unexposed women were 1.2 times more likely to have a molecular subtype associated with a better prognosis of disease compared to the exposed ones. The most prevalent molecular subtype in unexposed patients was Luminal A, which has slow-growing characteristics characterized by low rates of ki67 proliferation [65]. On the other hand, in pesticide-exposed women, we observed 1.5 times more likely to have the triple-negative molecular subtype, which characterized by more aggressive clinical behavi [66] and [67]. This trend towards a worse prognosis of patients with breast cancer and exposed to pesticides was also described in previous studies conducted by Pizzatti et al. (2020) [67] and Scandolara et al. (2022) [3], corroborating that pesticide-exposed women may have a poor disease prognosis.

In the exploratory analysis, a higher prevalence of patients with the Luminal B molecular subtype (34.37%) was identified, followed by a minimal difference between Luminal A patients (33.24%). This small difference in frequency between

the Luminal subtypes found in the study population differs from the percentage found in the populations of China (65.30% Luminal A and 19% Luminal B) [68] and the United States (55% Luminal A and 17% Luminal B) [69], countries that are among world leaders in the use of pesticides along with Brazil [70]. Although the population of these countries faces similar pesticide exposure as the Brazilian population, one probable reason why do he have a prevalence of more aggressive disease is that the studied population belongs to a region characterized by family farming. This type of exposure is chronic and severe, because people are continuously handling pesticides and contaminated items without any protection equipment. This contrasts with other countries where mechanized agriculture is more prevalent, reducing direct human contact with hazardous substances. This information towards to the emergence of proliferative tumors with molecular subtypes with unfavorable prognosis by pesticide exposure [71,72].

Pesticide exposure is known to be associated with immune dysregulation and inflammatory responses [73]. The study conducted by Silva et al. (2022) [9] identified a predominance of intermediate risk for death and recurrence in women exposed to pesticides, characterized by the prevalence of Luminal B tumors in association intermediate size tumors (between 2 and 5 cm) and intermediate tumor grade. In such patients, disease recurrence can occur due to the lack of treatment or a systemic damage can be induced by excessive treatment. Pesticide exposed patients categorized as intermediate risk also have poor immunological profiles, leading to unfavorable outcomes. Other studies have reported immune deregulation in women exposed to pesticides [44,71], which may favor the development of more aggressive tumors. This dysregulation may have triggered the higher frequency of distant metastases and recurrence observed in patients occupationally exposed to pesticides in this study.

The study developed by Scandolara et al. (2022) [3] investigated the impact of pesticide exposure on the mutational landscape in genes involved in homologous recombination and damage response in breast cancer patients. By analyzing tumor samples from 158 patients, it was found that pesticide exposure was associated with more pathogenic mutations, particularly in those diagnosed before the age of 50 or carrying variants in *BRCA1*, *BRCA2* or *PALB2*. These findings suggest that pesticide exposure may impact cancer development, mutational burden and disease progression.

Luminal B breast cancer patients exposed to pesticides were more likely to develop lymph node invasion and distant metastasis than other subtypes. Comparatively, we observed an increased frequency when compared to samples from other molecular subtypes. Li et al. (2019) [74] suggested that lymph node involvement and tumor metastasis have a strong association with pathogenic alterations in TP53 expression. These modifications were observed with a higher mutagenic frequency in patients exposed to pesticides in the study conducted by Scandolara et al. (2022) [3], demonstrating the cascade of progressive damage generated by pesticide exposure, including oncogenesis due to DNA impairment.

A study conducted on a population chronically exposed to pesticides at low exposure doses found cumulative DNA lesions due to failures in the genetic repair system [75]. Thus, the accumulation of mutations observed in the immune and inflammatory responses triggered by pesticide exposure [3,44,67,76], which generally result in the inactivation of tumor suppressor genes, are indications of genomic instability [77]. This genomic instability increases susceptibility to metastasis development [78], corroborating with 1.4 times more chances of patients exposed to pesticides to develop metastases.

Song et al. (2011) [79] found a greater predisposition to lymph node involvement in patients with angiolymphatic emboli, indicating a greater tendency to a worse prognosis. However, studies on the influence of the presence of angiolymphatic emboli on breast cancer are still scarce. In addition, due to the hypoxic environment that the presence of angiolymphatic emboli provides, some authors consider it as a precursor in the development of metastasis in cancer patients [73,80,81], reinforcing the idea of a greater tendency to an unfavorable prognosis for the patient, especially in the context of pesticide exposure.

Pizzatti et al. (2020) [67] reported that pesticide-exposed patients in menopause had significantly reduced levels of tumor necrosis factor-alpha (*TNF-α*) when compared to unexposed patients, suggesting that exposure may affect the production of *TNF-α* in the absence of estrogen, resulting in worsening of the disease due to the failure of antitumor mechanisms. This could help to explain the higher incidence of the triple-negative molecular subtype in the population exposed to pesticides.

This study has limitations, including modest sample size and the lack of other risk factors, such as dietary habits and lifestyle. Its strength is that the correlational analysis of pesticide exposure with clinicopathological parameters of breast cancer may be influencing the worse prognosis found in patients living in southwestern Paraná. It is important to remember that observational correlation does not necessarily imply causality, as a limitation of any exploratory work.

These findings are clinically relevant since we demonstrated the urgent need for targeted healthcare strategies for women exposed to pesticides, particularly in rural areas with high pesticide use such as Brazil. Our data suggest that pesticide exposure may be linked to more aggressive forms of breast cancer, with worse prognoses including increased recurrence, chemoresistance, and metastasis. These results highlight the importance of early detection and personalized treatment approaches for pesticide-exposed individuals, potentially guiding clinicians to monitor for aggressive disease progression and tailor therapeutic regimens accordingly.

In conclusion, although patients occupationally exposed to pesticides had a prevalence of the Luminal B subtype (33.33%), which is a molecular subtype commonly associated with a milder pathology prognosis compared to other molecular subtypes, these women had a higher frequency of disease recurrence (10.19%), chemoresistance (21.26%), and death (7.21%) than patients unexposed to pesticides. In addition, these women were also more likely to have distant metastases (1.4 times) and lymph node invasion (1.3 times) compared to patients unexposed to pesticides with breast cancer, indicating greater aggressiveness in the development of the disease. Given these findings, we reiterate the urgency of discussing and changing policies that regulate the use of pesticides and the need to screen exposed populations and those at risk of developing more aggressive diseases.

## Supporting information

**S1 File. S1 Table.** Association between molecular subtypes and clinical pathological variables with significance in breast cancer patients occupationally exposed to pesticides. **S2 Table.** Association between molecular subtypes and clinical pathological variables with significance in breast cancer patients unexposed to pesticides.
(ZIP)

## Acknowledgments

The authors are grateful to Carlos Chagas Institute, Fiocruz/PR and the CEONC staff for their excellent technical assistance and the patients who volunteered for the study.

## Author contributions

**Conceptualization:** Isabella C. Cazagranda, Maria Paula de Andrade Berny, Carolina Coradi, Daniel Rech, Carolina Panis, Guilherme F. Silveira.

**Data curation:** Isabella C. Cazagranda, Rafaela Frederico de Almeida, Lucca L. Smaniotto, Maria Paula de Andrade Berny, Carolina Coradi, Daniel Rech, Carolina Panis, Guilherme F. Silveira.

**Formal analysis:** Carolina Panis, Guilherme F. Silveira.

**Investigation:** Carolina Panis.

**Project administration:** Carolina Panis, Guilherme F. Silveira.

**Supervision:** Carolina Panis, Guilherme F. Silveira.

**Writing – original draft:** Isabella C. Cazagranda, Rafaela Frederico de Almeida, Lucca L. Smaniotto, Maria Paula de Andrade Berny, Carolina Coradi, Daniel Rech, Carolina Panis, Guilherme F. Silveira.

**Writing – review & editing:** Isabella C. Cazagranda, Rafaela Frederico de Almeida, Lucca L. Smaniotto, Maria Paula de Andrade Berny, Carolina Coradi, Daniel Rech, Carolina Panis, Guilherme F. Silveira.

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
