## [Decision Letter · Decision Letter 0]

5 Feb 2025

Dear Dr. Silveira,

Thank you for submitting your manuscript to PLOS ONE. After careful consideration, we feel that it has merit but does not fully meet PLOS ONE’s publication criteria as it currently stands. Therefore, we invite you to submit a revised version of the manuscript that addresses the points raised during the review process.

We look forward to receiving your revised manuscript.

Kind regards,

Elingarami Sauli, PhD

Academic Editor

PLOS ONE

Journal Requirements:

“The author(s) declare that financial support was received for the research, authorship, and/or publication of this article. Part of this research was supported by Carlos Chagas Institute, Fiocruz/PR, Fundação Araucária (grant number 639/2022) and by Conselho Nacional de Desenvolvimento Científico e Tecnológico – CNPq (grants number 402364/2021-0, 441017/2023-1, and 305335/2021-9)”

Additional Editor Comments (if provided):

**Comments from the Journal Office:**

Upon internal evaluation of the reviews provided, we kindly request you to disregard the reviewer report provided by Reviewer 3. No amendments are required in response to reviewer 3’s comments’

Reviewers' comments:

Reviewer's Responses to Questions

**Comments to the Author**

1. Is the manuscript technically sound, and do the data support the conclusions?

Reviewer #1: Partly

Reviewer #2: Yes

Reviewer #4: No

Reviewer #5: Yes

Reviewer #6: Partly

2. Has the statistical analysis been performed appropriately and rigorously?

Reviewer #1: No

Reviewer #2: Yes

Reviewer #4: No

Reviewer #5: Yes

Reviewer #6: Yes

3. Have the authors made all data underlying the findings in their manuscript fully available?

Reviewer #1: Yes

Reviewer #2: Yes

Reviewer #4: No

Reviewer #5: Yes

Reviewer #6: Yes

4. Is the manuscript presented in an intelligible fashion and written in standard English?

Reviewer #1: Yes

Reviewer #2: Yes

Reviewer #4: No

Reviewer #5: Yes

Reviewer #6: Yes

Reviewer #1: Dear authors,

The article is good, but some explanations are needed.

The main problem is numerical errors. The numbers in the tables and in the text are different. The main conclusion is not consistent with the numerical values presented.

I suggest a review of the statistical analysis performed. Below are the main questions, requests and suggestions.

Important questions.

Question 1 – The study article is about the existence of a statistical correlation between pesticides and breast cancer. There is a sample of 386 patients with breast cancer, of which 208 patients were exposed to pesticides, while 141 patients were not exposed to pesticides. In this count is missing 37 patients, probably without this information (missing value).

Line 206-207: “..After excluding patients with benign lesions, 386 patients

207 having a breast cancer diagnosis were included in the study..”

Line 226-227: “..Based on this, patients were categorized as occupationally exposed (n = 208) or unexposed to pesticides (n = 141).

Exposed patients (n = 208) + Unexposed patients (141) = 349 patients ≠ Total of patients (386). It is necessary to explain this mathematics.

Question 2 – Lines 88-90. “..A total of 386 women were included in the study. After a structured interview, women were categorized as exposed (n=208) or unexposed (n=141) to pesticides..”

If this study is about whether there is a statistical correlation between pesticides and breast cancer, shouldn't it only consider the 349 patients classified as exposed or not exposed to pesticides?

Question 3 – If this study is about whether there is a statistical correlation between pesticides and breast cancer, why consider patients about whom we do not have this information?

What is the purpose of including these 37 patients without classification regarding pesticide exposure?

Question 4 – Line 307: “..the Luminal B molecular subtype was prevalent (32.83%),..”

Luminal B molecular subtype values in Table 1 (33.33%) and in line 307 (32.83%) are different. Furthermore, the Luminal B molecular subtype was NOT prevalent in Table 1. Explain.

Question 5 – Problem: Values in Table 1 ≠ values in text. Correct the values presented

Line 299: ‘..33.8% as Luminal B,..” Table 1 � 34.37%

Line 300: “..16.34% as Triple-Negative..” Table 1 � 15.77%

Check the other numbers.

Question 6 – line 368-369. Statement inconsistent with the values presented in Table 1.

“..and predominance of the molecular subtype Luminal B..”

Question 7 - line 383-384: Correct the values presented

“..In the exploratory analysis, a higher prevalence of patients with the Luminal B molecular subtype (33.8%) was identified..”

General questions

1 – Check in PLOS ONE whether reference citations should be placed before or after the punctuation marks. Example: “..carcinogenic potential,3,4 ..” or “..carcinogenic potential3,4,..”

Other questions, requests, and suggestions.

Line 101 . Deleted: to breast cancer

“..compared to patients not exposed to breast cancer. Conclusions: These findings..”

Line 116-117. Normalmente a citação de referência pertence ao texto. Verifique o estilo da PLOS ONE.

“..carcinogenic potential,3,4 endocrine disrupting properties,5 genotoxicity, 6–8 and immunotoxicity.9–12,..”

Suggestion: Comma after reference

“..carcinogenic potential3,4, endocrine disrupting properties5, genotoxicity6–8, and immunotoxicity9–12,..”

Line 117. Change “..immunotoxicity.9–12,..” to “immunotoxicity,9–12.. or immunotoxicity9–12,..”

Line 125. Change “..as the European Union..” to “..as in the European Union..”

Line 126-127. Deleted: of women

“..where they represent 45 % of women, and certain regions..”

Line 129. Delete: breast

“..the incidence of breast cancer in women..”

Reviewer #2: Dear authors,

The results described in the article relate to epidemiological studies and are extremely relevant.

There is no doubt that the effect of pesticides is reflected in the genetic status of the patient. Some mutations appear. I leave it to the authors' decision whether to include data on the characteristics and presence of some mutations specific to the region, or the presence of ethnic mutations. This would be very interesting.

Comments:

1. Figure 1 lists all 27 municipalities. A similar list of these municipalities appears in lines 188-199. You need to remove this repetition from the text of the article.

2. Subsection 2.1.2. Tumor characteristics, line 230, should be shortened, since all the data are given in the table.

Reviewer #4: This is a study of risks associated with occupational pesticide exposure. It includes 386 breast cancer patients from a region in Brazil known for high use of pesticides. A structured interview was performed, and clinical and pathology data were drawn from medical records. The authors conclude that pesticide exposure favors the occurrence of more aggressive breast cancer.

It is important to study the risks associated with pesticide use in agriculture among women in Brazil. Nevertheless, I have concerns regarding the work presented in this study.

My main concern is the number of patients included, with only 208 exposed, and 141 unexposed patients.

Introduction:

The introduction explains the rationale for studying the association between pesticide use and breast cancer pathology.

Methods:

It says that patients were recruited from 2015 to 2017 (page 4, line 178). In the same page, line 182 it seems like the recruitment period was from 2015 to 2023.

It says that “this information could not identify individual participants” (page 4, line 178-179). What does that mean? Since medical records were used, individual patients would need to be identified.

It is not necessary to name all the municipalities (page 4).

Due to the high use of pesticides in this region in Brazil, are the authors sure that the unexposed group are truly unexposed to pesticides? The definition of “exposed” includes only highly exposed patients (living at least 50% of their lives under direct handling of such substances at least twice a week for every week of the year). By using such a strict definition, I would assume that you would miss many of the pesticide exposed patients. This needs to be addressed in the discussion. Starting with 389 patients, only 349 (208 + 141) seem to be included. Yet, looking at table 1, n=353.

With regards to tumor characteristics, section 2.1.2: The authors need to describe more thoroughly regarding the biomarkers. For instance, ER and PR are probably assessed by immunohistochemistry (IHC). It is stated that ER and PR >0 is defined as positive, but according to ASCO/CAP guidelines, a cut-off of 1% is used. Regarding HER2 it says “values of 3+…” This refers to HER2 protein expression assessed by IHC. The authors must separate between gene copy number (amplification), assessed by in situ hybridization, and protein expression, assessed by IHC. The choice of cut off (14%) for Ki67 should also be discussed.

With regards to disease characteristics: The risk stratification for death and recurrence should be explained, not only refer to a paper. What was the follow up from diagnosis? How was chemoresistance defined?

Results:

Heading 3.1. “Breast cancer patients occupationally exposed to pesticides have a prevalence of the Luminal B molecular subtype”. This sentence does not make sense.

A table should not include only statistically significant results. It is stated that “other statistical values analyzed are present in the supplementary materials”.

The way the results are presented makes it very hard for the reader to follow. Example: “Increased frequency of Luminal B tumors was observed in relation to HER2-amplified (p=<0,01 and ratio= 2.31) and Triple-negative (p=<0,01 and ratio 2,73) in the exposed group (Table 2) regarding distant metastasis. So, in one sentence, molecular subtype, exposure status and distant metastasis are addressed. The same applies to the next sentence, and the way the tables are organized.

In Figure 1, the cases that are neither exposed nor unexposed are not presented. Not all 389 patients are included (208 + 141=349).

In the tables, the numbers don’t add up, and some numbers are missing. Example: n=353 in Table 1, not 349. It says in the subtext that some variables are missing, but when only the “positive” number is given, it is not possible for the reader to see the number of missing values. For example, both ER positive and ER negative should be listed in the table. Why is age missing? When data is collected from medical records, I would assume age is available for all. The distribution of molecular subtypes varies depending on age. Whether the cancer was clinical or detected by mammography could also affect the distribution of molecular subtypes. This information is missing.

Discussion:

Strengths and weaknesses should be discussed more thoroughly. The strength mentioned in page 13, line 452-455 is not really a strength.

Last paragraph, page 12: This is more introduction-like.

The paragraph in page 13, line 444-450: The conclusion is not sound.

Conclusion:

It is not true that the Luminal B subtype is associated with a good prognosis (page 13, line 457-458).

General comments:

Several sentences are not precise enough. For example: “This trend towards a worse prognosis of women….” (page 11, line 379-380). Breast cancer must be mentioned for this sentence to make sense.

The manuscript needs to be revised due to several grammatical errors and spelling mistakes. Example: “The patients were recruitment….” (page 4, line 176-177) and “…, one probable reason why do he have a prevalence…”. (page 11, line 391).

It is important to distinguish associations and causality. An association does not necessarily mean that there is causality. This is very important and must be addressed more thoroughly.

Reviewer #5: I have gone through the manuscript titled "Hidden risks associated with occupational pesticide exposure in women with breast cancer: high frequency of the Luminal B molecular subtype and occurrence of poor prognostic features".

I have the following observations regarding the manuscript:

1.The study lacks some clinical applications perspectives. Although the study identifies an association between pesticide exposure and breast cancer, it does not adequately discuss the causal mechanisms between exposure and tumor progression or immune dysregulation.

2.The current exposure standard requires at least two exposures per week, every week of the year, which is a broad criterion. It does not thoroughly consider that different types of pesticides (e.g., carcinogenic or endocrine-disrupting) may have different health impacts, particularly the effects of exposure dose and exposure type.

3.There is a lack of long-term follow-up data. It is recommended to increase the sample size and conduct long-term follow-up studies to better understand the long-term effects of pesticide exposure.

Based on these observations, I would like to give a 'major revision' of the manuscript as decision. All the best to the authors!

Regards

Reviewer #6: The manuscript "Hidden risks associated with occupational pesticide exposure in women with breast cancer: high frequency of the Luminal B molecular subtype and occurrence of poor prognostic features" investigated the impact of chronic and continued occupational pesticide exposure on the clinicopathological profiling of breast cancer, with the aim of determining a clinicopathological signature associated with pesticide exposure.

This study presents a better comprehension about the pesticides harmful in relation to the clinicopathological characteristics of tumor, as molecular subtype, grade, average tumor size, presence of angiolymphatic emboli, lymph node positivity and distant metastasis, besides the characteristics of the disease, as stratification of the risk of death and recurrence, recurrence profile, chemoresistance to treatment and occurrence of death. In this sense, the topic investigated in the study is relevant to environmental, health and oncology. Furthermore, this is a clinical study that can help medical decisions about the best treatment protocols when considering the patient exposition or not to pesticides. Finally, the study is important to alert the population about the dangers and impacts caused by pesticides and to assist public health in the sense of prevention.

The introduction shows clearly the importance to study pesticides and breast cancer e also presents the novelty and the objective of the study. Besides, methods section is described satisfactorily and presents all necessary points to the study comprehension. Table 1 reinforce every method applied in the study. The discussion also demonstrates justificatives of the results and comparison with the literature. However, results section need authors attention, because the data showed have not mathematical concordance and need major review.

In view to improve the manuscript quality some appointments are listed below:

1. Keywords: the words already present in the title should be changed by other words, in view of increase the possibilities of manuscript search. For example: pesticides by agrochemicals and breast cancer by breast tumor or breast neoplasm.

2. Line 186 in the number 500,000 inhabitants the comma need be changed by point.

3. Table 1 subtitle - the following municipalities should maintain the original language of the name: 16 - Pérola D' west -> Pérola D'Oeste; 20 - Royalty -> Realeza; 21 - Renaissance - Renascença.

4. Table 1 data and their results presented in the text need major review. For examples:

- Total population presented is 386, but the population exposed is 208 and not exposed is 141, however, 208 + 141 = 349 and not 386.

- Mean age of 353 patients – 197 exposed and 134 not exposed, but 197 + 134 = 331 and not 353.

- Menopausal status of 229 patients – 130 exposed and 89 not exposed, but 130 + 89 = 219.

- BMI of 259 patients – 150 exposed and 97 not exposed, but 150 + 97 = 247.

- Luminal A 33.24% (n=118) 30.81% (n=61) 37.78% (n=51), but 61 + 51 = 112 and not 118.

- Luminal B 34.37% (n=122) 33.33% (n=66) 34.81% (n=47), but 66 + 47 = 113 and not 122.

- HER2-amplified 16.62% (n=59) 16.67% (n=33) 15.56% (n=21), but 33 + 21 = 54 and not 59.

- Triple-negative 15.77% (n=56) 19.19% (n=38) 11.85% (n=16), but 38 + 16 = 54 and not 56.

- Presence of angiolymphatic emboli 25.52% (n=86) 27.51% (n=52) 24.06% (n=32), but 52 + 32 = 84 and not 86.

… other...

If there is a equivocate mathematical it need be adjusted and if there is not this difference need be explicated. Besides, the percent of these data also may be equivocated, since the number of exposed and not exposed patients would be not correct.

5. The other next tables may present data also equivocate, in view of the difference in the calculi, as described above. Moreover, the abstract results, the results section and the discussion of the results also be compromited.

6. NA described in tables 3 and 4 needs to be presented in full at their respectives subtitles.

7. Insert p value in table 1 in order to better comprehension of the table, without necessary to read complementary text results.

**Do you want your identity to be public for this peer review?** For information about this choice, including consent withdrawal, please see our Privacy Policy

Reviewer #1: **Yes:** Paulo Laerte Natti

Reviewer #2: No

Reviewer #4: No

Reviewer #5: No

Reviewer #6: No

---

## [Author Response · Author response to Decision Letter 1]

19 Feb 2025

The response to the reviewers was added to the document along with the revised article.

---

## [Decision Letter · Decision Letter 1]

28 Mar 2025

Dear Dr.  Silveira,

Thank you for submitting your manuscript to PLOS ONE. After careful consideration, we feel that it has merit but does not fully meet PLOS ONE’s publication criteria as it currently stands. Therefore, we invite you to submit a revised version of the manuscript that addresses the points raised during the review process.

Please submit your revised manuscript by May 12 2025 11:59PM. If you will need more time than this to complete your revisions, please reply to this message or contact the journal office at plosone@plos.org . A rebuttal letter that responds to each point raised by the academic editor and reviewer(s). You should upload this letter as a separate file labeled 'Response to Reviewers'.A marked-up copy of your manuscript that highlights changes made to the original version. You should upload this as a separate file labeled 'Revised Manuscript with Track Changes'.An unmarked version of your revised paper without tracked changes. You should upload this as a separate file labeled 'Manuscript'.

We look forward to receiving your revised manuscript.

Kind regards,

Elingarami Sauli, PhD

Academic Editor

PLOS ONE

Journal Requirements:

Additional Editor Comments :

The authors should address all the reviewer comments, including highlighting the limitations of the study in their conclusion.

Reviewers' comments:

Reviewer's Responses to Questions

**Comments to the Author**

Reviewer #1: All comments have been addressed

Reviewer #7: All comments have been addressed

2. Is the manuscript technically sound, and do the data support the conclusions?

Reviewer #1: (No Response)

Reviewer #7: Yes

3. Has the statistical analysis been performed appropriately and rigorously?

Reviewer #1: (No Response)

Reviewer #7: Yes

4. Have the authors made all data underlying the findings in their manuscript fully available?

Reviewer #1: (No Response)

Reviewer #7: Yes

5. Is the manuscript presented in an intelligible fashion and written in standard English?

Reviewer #1: (No Response)

Reviewer #7: Yes

Reviewer #1: All questions were addressed and answered appropriately.

The research data are available in the text and tables of the article.

Reviewer #7: (No Response)

**Do you want your identity to be public for this peer review?** For information about this choice, including consent withdrawal, please see our Privacy Policy

Reviewer #1: **Yes:** Paulo Laerte Natti

Reviewer #7: No

---

## [Author Response · Author response to Decision Letter 2]

17 Apr 2025

A response letter addressing all the comments made by the reviewers and the editor is being submitted.

---

## [Decision Letter · Decision Letter 2]

15 Aug 2025

Dear Dr. Silveira

Thank you for submitting your manuscript to PLOS ONE. After careful consideration, we feel that it has merit but does not fully meet PLOS ONE’s publication criteria as it currently stands. Therefore, we invite you to submit a revised version of the manuscript that addresses the points raised during the review process.

We look forward to receiving your revised manuscript.

Kind regards,

Elingarami Sauli, PhD

Academic Editor

PLOS ONE

Journal Requirements:

Reviewers' comments:

Reviewer's Responses to Questions

**Comments to the Author**

Reviewer #1: All comments have been addressed

Reviewer #7: All comments have been addressed

Reviewer #8: (No Response)

Reviewer #9: (No Response)

2. Is the manuscript technically sound, and do the data support the conclusions?

Reviewer #1: (No Response)

Reviewer #7: Yes

Reviewer #8: Yes

Reviewer #9: No

3. Has the statistical analysis been performed appropriately and rigorously?

Reviewer #1: (No Response)

Reviewer #7: Yes

Reviewer #8: No

Reviewer #9: No

4. Have the authors made all data underlying the findings in their manuscript fully available?

Reviewer #1: (No Response)

Reviewer #7: Yes

Reviewer #8: No

Reviewer #9: No

5. Is the manuscript presented in an intelligible fashion and written in standard English?

Reviewer #1: (No Response)

Reviewer #7: Yes

Reviewer #8: Yes

Reviewer #9: No

Reviewer #1: (No Response)

Reviewer #7: The authors have satisfactorily addressed all my comments/suggestions. Since in the first review I proposed only a "minor revision", I would like to point out that the manuscript is now ready for publication.

Reviewer #8: The manuscript "Hidden risks associated with occupational pesticide exposure in women with breast cancer: high frequency of the Luminal B molecular subtype and occurrence of poor prognostic features" presents a study of how breast cancer subtypes distributes changes between exposed / non-exposed pesticide groups.

The article is clearly written and the analysis direct, and in a very relevant topic.

My main comment is if there are not confounding variables, for instance education level / subject income / ancestry ? I'm worried exposure to pesticide could be highly correlated to some other variables.

If the authors have some data about that will be great, if not at least should be mentioned as a limitation.

Some minor comments:

- Please check "2,4-D?"

- Can you detail "Inconsistencies in the data?"

Reviewer #9: Hidden risks associated with occupational pesticide exposure in women with breastcancer: high frequency of the Luminal B molecular subtype and occurrence of poor prognostic features.

Title: Luminal B.

We observed limitations of information related to luminal B.

We have luminal B her negative and luminal B her positive.

Review it.

Methods

1. Improve localization

Francisco Beltrão Cancer Hospital (Ceonc) an Hospital Cancer Center that serves patients from western Parana Stage.

2. Organize the information

- Images suggestive of breast lesions identified by mammograms and ultrasound were included

Evaluate breast lesions? Birads IV, V or Birads III, IV or V?

- Imagens suggestive of adnominal breast lesions (Birads IV or V ?) identified by mammograms and ultrasound were included

- Biopsy of suspicious lesion = Birads IV or V

3. inconsistencies in the data?

Related to the condition?

37/386 inconsistencies = 9.5%

37 (9.5%) of inconsistency – inform it

4. comprehensive questionnaire with 61 questions

Consider it in supplementary material

If you used the 60 questions by the Thesis of Shaiane Carla Gaboardi, consider it methods

https://sucupira.capes.gov.br/sucupira/public/consultas/coleta/trabalhoConclusao/viewTrab alhoConclusao.jsf?popup=true&id_trabalho=10956083

5. Report a table showing all conditions related to the group considered exposed to pesticide

6. Despite similar geographic locations, genetic backgrounds, and cultural and dietary habits, the unexposed group had no occupational pesticide exposure.

genetic backgrounds and cultural habits

All patients were born in this region?

Do you have information about race?

If no, delete this information

7. Tumor characteristics

- Ann Oncol. 2023; Nov 1;34(11):970–86. The reference do not classify the tumors

- The publication is: Strategies for subtypes—dealing with the diversity of breast cancer: highlights of the St Gallen International Expert Consensus on the Primary Therapy of Early

Breast Cancer 2011. Annals of Oncology 2011; 22: 1736-1747

8. Based on St Gallen consensus we have Luminal B Her positive.

You do not reported these patients?

He have: Luminal A, Luminal B her negative, Luminal B her positive, Triple negative and Her2 (triple positive)

9. Patients were from 2015 to 2023. To evaluate recurrence the cutoff time must be in 2018 (five years).

To evaluate these conditions, we must have time and Kaplan Meier methods.

- Do not evaluate recurrence or survival. To evaluate this conditions, you have to report other informations as disease free-survival, cancer specific survival, mean follow up, status of the patients at the end of study.

- Consider a transversal study evaluating patients at diagnosis. No not evaluate death.

10. Disease characteristics

- Chemoresistance development: You considered RECIST

Patients were submitted to chemotherapy

Inform number or patients, type of chemotherapy and disease response

-Stratification for death and recurrence reference 59

The data is inconsistent This page no longer exists. This is an order to release APAC.

If you want to report chemoresistance, report information about tumor characteristics, chemotherapy and response. If no exclude this information

Results

- Your study appears to be well-done, but the characteristics of the patients and tumors are not reported, which prevents us from assessing the study's reproducibility. Also the criteria used to consider exposed/ unexposed.

- Please include this information in a supplementary table.

- Clinical characteristics at diagnosis: Age, race, tumor mean size, TNM, T-TNM, N-TNM, M-TNM, molecular subtype (5 types),

- Treatment: neoadjuvant chemotherapy, response to chemotherapy; type of treatment, follow up, disease free-survival, sites of recurrence, overall specific survival, status at last follow up

Improve Table one

Variable – caractheristics – exposed – unexposed – total – p

Organize variables. For example, molecular subtype is one variable; grade is one variable.

Disease characteristics – inconsistency data

11- What the criteria used?

Of this population, 8.25% were stratified as low risk for 304 death and recurrence, 55.87% into intermediate risk, and 35.87% were classified 305 as high risk.

12- Review molecular evaluation/ Luminal B Her positive/ Luminal B her negative

Regarding the molecular subtype, 33.24% of the patients were

classified as Luminal A, 34.37% as Luminal B, 16.62% as HER2-amplified, and

15.77% as Triple-Negative.

13. Follow Up? Condition of the patient at last follow up?

About 7% of the patients died, 9.36%of the patients had disease recurrence

14. Chemoresistance? At neoadjuvant chemotherapy, disease progression after chemotherapy?

18.97% of the patients developed chemoresistance.

15. Luminal A versus Luminal B?

Luminal B her positive?

16. Discussion is based of results. Review the methods and results

Conclusion:

Consider major modifications or rejection

**Do you want your identity to be public for this peer review?** For information about this choice, including consent withdrawal, please see our Privacy Policy

Reviewer #1: **Yes:** Paulo Laerte Natti

Reviewer #7: No

Reviewer #8: No

Reviewer #9: No

---

## [Author Response · Author response to Decision Letter 3]

3 Sep 2025

Dear Dr. Sauli.

We would like to thank you for your analysis, criticism and considerations, as well as those of the reviewers, regarding our work for submission to this prestigious journal. Below are the responses to each point raised which, we hope, can address all criticisms of the manuscript.

---

## [Decision Letter · Decision Letter 3]

23 Sep 2025

Dear Dr. Silveira,

Thank you for submitting your manuscript to PLOS ONE. After careful consideration, we feel that it has merit but does not fully meet PLOS ONE’s publication criteria as it currently stands. Therefore, we invite you to submit a revised version of the manuscript that addresses the points raised during the review process.

**ACADEMIC EDITOR:**

We look forward to receiving your revised manuscript.

Kind regards,

Elingarami Sauli, PhD

Academic Editor

PLOS ONE

Journal Requirements:

Reviewers' comments:

Reviewer's Responses to Questions

**Comments to the Author**

Reviewer #8: All comments have been addressed

Reviewer #9: (No Response)

2. Is the manuscript technically sound, and do the data support the conclusions?

Reviewer #8: Yes

Reviewer #9: No

3. Has the statistical analysis been performed appropriately and rigorously?

Reviewer #8: Yes

Reviewer #9: No

4. Have the authors made all data underlying the findings in their manuscript fully available?

Reviewer #8: Yes

Reviewer #9: No

5. Is the manuscript presented in an intelligible fashion and written in standard English?

Reviewer #8: Yes

Reviewer #9: Yes

Reviewer #8: All comments have been addressed satisfactory by the authors including limitations and further details on methods, no more comments from my side.

Reviewer #9: - Positive point: The definition of pesticide-exposed and non-exposed individuals is described in a previous study.

-However, the study has methodological flaws that need improvement. These were already reported in the first evaluation, but no improvements were made. I therefore confirm the observations.

-Molecular classification. St. Gallen. Reference 58. There are four intrinsic subtypes, but five molecular subtypes. Please separate Luminal B into Luminal B her-positive and Luminal B her-negative, as they have different treatment and prognosis. Studies must answer clinical questions, making it necessary to separate the five molecular subtypes.

- Use the RECIST criteria for breast cancer. The RECIST criteria refer to the breast, but lymph node disease also exists in breast cancer. Please add the NSABP criteria, where a complete pathological response corresponds to the absence of invasive disease in the breast and axilla.

-Disease characteristics:

(1) Patients with early-stage breast cancer who underwent neoadjuvant chemotherapy and were subsequently followed for disease recurrence were included in the study.

Correct information: All patients were followed, or the rate of lost of follow up.

(2) Individuals who developed recurrence or systemic progression received adjuvant therapy. Correct information: Adjuvant therapy is performed after surgical treatment. In the presence of recurrence or metastasis, treatment is palliative.

- The study reports that assessed prognosis. Prognosis depends on recurrence and death. Follow-up time is important. In this context, we must consider variables such as follow-up, recurrence, death from cancer, death from other causes, and follow-up time. These are minimal variables for comparing studies, as well as the actuarial survival curve.

-The study describes recurrence at 10.19% and death at 7.21%. These data are not from Brazil, where I live, and I work with breast cancer. Low numbers of these are possibly related to short follow-up or loss to follow-up.

-The study describes that the last patient was admitted in 2023. There was not enough time for recurrence and death for all patients. Follow-up time is variable, with missing data, constituting a limitation of the study.

- Table 1 lacks a description of the numerical evaluation of the differences between the groups. For example: Estrogen receptor positive/negative and p. It would require an extra row and a column with all the "p"s.

-Separate continuous variables from categorical ones. For continuous variables, present the standard deviation. Consider five molecular subtypes.Basic variables missing from any breast cancer study, such as T-N-M stage, are missing.

-Risk and recurrence stratification is based on reference 59. The page no longer exists. Please replace it with a scientific article, not a government website.

-The limitations in presenting these data compromise the results in Table 2.

-The limitations in presenting basic situations such as T-N-M, molecular subtype divided into 5 categories, pathological response classification, recurrence and death rates with long follow-up and low loss to follow-up.

In conclusion: Due to these conditions associated with the method, which have not been corrected in this version, I do not recommend acceptance of the article.

**Do you want your identity to be public for this peer review?** For information about this choice, including consent withdrawal, please see our Privacy Policy

Reviewer #8: No

Reviewer #9: No

---

## [Author Response · Author response to Decision Letter 4]

27 Sep 2025

Dear Dr. Sauli.

We would like to thank you again for your analysis, criticism and considerations. In the point-by-point response we include the information requested by the editor.

---

## [Decision Letter · Decision Letter 4]

22 Oct 2025

Dear Dr. Silveira,

Thank you for submitting your manuscript to PLOS ONE. After careful consideration, we feel that it has merit but does not fully meet PLOS ONE’s publication criteria as it currently stands. Therefore, we invite you to submit a revised version of the manuscript that addresses the points raised during the review process.

The authors should ASAP address raised methodological issues....or at least list them as limitations (in detail) if they can't address them.

We look forward to receiving your revised manuscript.

Kind regards,

Elingarami Sauli, PhD

Academic Editor

PLOS ONE

Journal Requirements:

Reviewers' comments:

Reviewer's Responses to Questions

**Comments to the Author**

Reviewer #9: (No Response)

2. Is the manuscript technically sound, and do the data support the conclusions?

Reviewer #9: No

3. Has the statistical analysis been performed appropriately and rigorously?

Reviewer #9: No

4. Have the authors made all data underlying the findings in their manuscript fully available?

Reviewer #9: Yes

5. Is the manuscript presented in an intelligible fashion and written in standard English?

Reviewer #9: No

Reviewer #9: 1. If the authors have adequately addressed your comments raised in a previous round of review and you feel that this manuscript is now acceptable for publication, you may indicate that here to bypass the “Comments to the Author” section, enter your conflict of interest statement in the “Confidential to Editor” section, and submit your "Accept" recommendation.

Correct response: The authors did not adequately implement the previous suggestions

In the first review, I highlighted several points that needed modification, considering major revisions.

In the second version, the changes were not made, and the points that needed improvement were included, and the article was rejected due to inadequate data.

Now, in the third version, the suggested points were not changed, so I must maintain the rejection finding.

The problem is methodological. It lists four breast tumor subtypes, and we have five. It describes recurrence without describing adequate follow-up. Unaddressed methodological issues prevent the article from being accepted.

**Do you want your identity to be public for this peer review?** For information about this choice, including consent withdrawal, please see our Privacy Policy

Reviewer #9: No

---

## [Author Response · Author response to Decision Letter 5]

30 Oct 2025

PONE-D-24-40245R2

Hidden risks associated with occupational pesticide exposure in women with breast cancer: high frequency of the Luminal B molecular subtype and occurrence of poor prognostic features.

Dear Dr. Sauli.

We would like to thank you again for your analysis, criticism and considerations. Below are the requested modifications.

Editor comments:

The authors should ASAP address raised methodological issues....or at least list them as limitations (in detail) if they can't address them.

Reviewer #9:

1. If the authors have adequately addressed your comments raised in a previous round of review and you feel that this manuscript is now acceptable for publication, you may indicate that here to bypass the “Comments to the Author” section, enter your conflict of interest statement in the “Confidential to Editor” section, and submit your "Accept" recommendation.

Correct response: The authors did not adequately implement the previous suggestions

In the first review, I highlighted several points that needed modification, considering major revisions.

In the second version, the changes were not made, and the points that needed improvement were included, and the article was rejected due to inadequate data.

Now, in the third version, the suggested points were not changed, so I must maintain the rejection finding.

The problem is methodological. It lists four breast tumor subtypes, and we have five. It describes recurrence without describing adequate follow-up. Unaddressed methodological issues prevent the article from being accepted.

R.: We sincerely apologize for this. We understand that there are different evaluation criteria, both for patient follow-up and for the classification of molecular subtypes, the two main points indicated by the last reviewer of the paper. We believe that these criteria are gradually modified over the years by the scientific community and that, therefore, different interpretations are possible, especially in a complex pathology such as breast cancer. Thus, we agree that it is necessary to clearly and objectively include the criteria that were used in the study, which we have sought to do in this last revision. We also believe that the work has already been thoroughly evaluated by eight different reviewers and that the contribution of the last reviewer, even if they do not agree with our criteria, can bring more clarity to the current version of the article.

Details about the follow-up period and recurrence assessment have been added in the revised version of the manuscript.

Methods section , Pag 9 and 10

“Imaging reports from mammography, diagnostic ultrasound, breast MRI, CT, and PET-CT during follow-up were analyzed by clinicians in accordance with RECIST recommendations to establish baseline and post-treatment status. For each patient, the same imaging modality was applied at baseline and follow-up to ensure consistency.”

“Treatment response was categorized as follows: complete response, defined as the disappearance of all target lesions; partial response, defined as at least a 30% reduction in target lesion size without evidence of new lesions; progressive disease, defined as a minimum 20% increase in lesion size relative to baseline and/or the appearance of new lesions in the breast or distant organs; and stable disease, defined as insufficient shrinkage to meet partial response criteria and insufficient increase to meet progression criteria. Patients were monitored over a five-year follow-up period. Based on these criteria, patients were classified as responsive (complete response) or chemoresistant (partial response, progressive disease, or stable disease).”

“We used only the RECIST (Response Evaluation Criteria In Solid Tumors) criteria for the follow-up period. This wis a globally recognized standard for evaluating a patient's tumor response to treatment, used to assess whether tumors shrank, stayed the same, or grew. It was the standard used by our medical team for years, which made it impractical to apply another criterion retroactively. The low mortality rate observed in our study was because all patients were in Stage II. Stage II cancer was considered an early stage and was often curable with treatments like surgery, radiation, or chemotherapy, which was a reason for the lower mortality rate. Our study only considered deaths from cancer, and we did not have access to information on deaths from other causes.”

We have now included a detailed explanation of the separation of molecular subtypes as a study limitation, acknowledging the absence of Luminal-HER2 cases as a factor that may restrict the generalizability of our findings, as follows:

“In previous classifications, the Luminal B subtype of breast cancer was often defined as estrogen receptor (ER)-positive and HER2-amplified, meaning that tumors expressing concomitant hormone receptors and HER2 overexpression were categorized as Luminal B (HER2-positive). Currently, the Luminal B subtype is identified independently of HER2 status, being characterized instead by positive hormone receptor expression (ER and/or PR) combined with a high proliferative index (Ki-67 >14%), regardless of HER2 amplification. The initial design of our study did not aim to include patients with the Luminal-HER2 subtype due to the small number of cases in this subgroup; therefore, this subgroup was excluded from the final analysis to avoid biased interpretations based on an insufficient sample size. Since their data were not collected, they could not be included in the present study. The results were presented considering only the four main breast cancer subtypes, considering the St. Gallen International Expert Consensus (Figure XXX), which recognizes that molecular classification based on immunohistochemistry groups breast tumors into these four main categories. We believe this decision does not compromise the methodological rigor or the scientific merit of our study (Coates 2015, Burstein 2023).”

Table XXX - Definition of subtypes of breast cancer included in the study accordingly to the St. Gallen classification (Goldhirsch et al., 2013)

Subtypes of breast cancer ER and PR HER2 Ki67

Luminal A ER + and/or PR + negative Ki-67<14%

Luminal B ER+ and/or PR+ negative Ki-67≥14%

HER2 enriched ER-, PR- positive Any Ki-67

Triple negative ER-, PR- negative Any Ki-67

Legend: negative (-), positive (+). Luminal-HER2 patients (ER+, PR+, and HER2+) were not included in the study.

A. Goldhirsch, E.P. Winer, A.S. Coates, R.D. Gelber, M. Piccart-Gebhart, B. Thürlimann, H.-J. Senn, Kathy S. Albain, Fabrice André, Jonas Bergh, Hervé Bonnefoi, Denisse Bretel-Morales, Harold Burstein, Fatima Cardoso, Monica Castiglione-Gertsch, Alan S. Coates, Marco Colleoni, Alberto Costa, Giuseppe Curigliano, Nancy E. Davidson, Angelo Di Leo, Bent Ejlertsen, John F. Forbes, Richard D. Gelber, Michael Gnant, Aron Goldhirsch, Pamela Goodwin, Paul E. Goss, Jay R. Harris, Daniel F. Hayes, Clifford A. Hudis, James N. Ingle, Jacek Jassem, Zefei Jiang, Per Karlsson, Sibylle Loibl, Monica Morrow, Moise Namer, C. Kent Osborne, Ann H. Partridge, Frédérique Penault-Llorca, Charles M. Perou, Martine J. Piccart-Gebhart, Kathleen I. Pritchard, Emiel J.T. Rutgers, Felix Sedlmayer, Vladimir Semiglazov, Zhi-Ming Shao, Ian Smith, Beat Thürlimann, Masakazu Toi, Andrew Tutt, Michael Untch, Giuseppe Viale, Toru Watanabe, Nicholas Wilcken, Eric P. Winer, William C. Wood. Personalizing the treatment of women with early breast cancer: highlights of the St Gallen International Expert Consensus on the Primary Therapy of Early Breast Cancer 2013, Annals of Oncology, Volume 24, Issue 9, 2013, Pages 2206-2223, ISSN 0923-7534, https://doi.org/10.1093/annonc/mdt303.

Coates, A. S., Winer, E. P., Goldhirsch, A., et al. (2015). Tailoring therapies—improving the management of early breast cancer: St. Gallen International Expert Consensus on the Primary Therapy of Early Breast Cancer 2015. Annals of Oncology, 26(8), 1533–1546.

Burstein, H. J., Curigliano, G., Thürlimann, B., et al. (2023). Customizing therapies for women with early breast cancer: St. Gallen International Consensus Conference 2023. Annals of Oncology, 34(7), 573–592.

---

## [Decision Letter · Decision Letter 5]

28 Nov 2025

Dear Dr. Silveira,

Thank you for submitting your manuscript to PLOS ONE. After careful consideration, we feel that it has merit but does not fully meet PLOS ONE’s publication criteria as it currently stands. Therefore, we invite you to submit a revised version of the manuscript that addresses the points raised during the review process.

**THE AUTHORS SHOULD ASAP ADDRESS RAISED METHODOLOGICAL ISSUES (by Reviewer 9 AS LIMITATIONS, IF THEY CANT BE SOLVED AS QUERIED. THIS SHOULD BE DONE ASAP BEFORE THIS SUBMISSION CAN BE ACCEPTED.**

We look forward to receiving your revised manuscript.

Kind regards,

Elingarami Sauli, PhD

Academic Editor

PLOS ONE

Journal Requirements:

Reviewers' comments:

Reviewer's Responses to Questions

**Comments to the Author**

Reviewer #8: All comments have been addressed

2. Is the manuscript technically sound, and do the data support the conclusions?

Reviewer #8: Yes

3. Has the statistical analysis been performed appropriately and rigorously?

Reviewer #8: Yes

4. Have the authors made all data underlying the findings in their manuscript fully available?

Reviewer #8: Yes

5. Is the manuscript presented in an intelligible fashion and written in standard English?

Reviewer #8: Yes

Reviewer #8: All comments have been addressed in previous round of revision, main changes incorporated in this version are more detailed limitations.

**Do you want your identity to be public for this peer review?** For information about this choice, including consent withdrawal, please see our Privacy Policy

Reviewer #8: No

---

## [Author Response · Author response to Decision Letter 6]

4 Dec 2025

We would like to thank you again for your analysis, criticism and considerations. Below are the requested modifications.

---

## [Editor Report · Decision Letter 6]

8 Dec 2025

Hidden risks associated with occupational pesticide exposure in women with breast cancer: high frequency of the Luminal B molecular subtype and occurrence of poor prognostic features.

PONE-D-24-40245R6

Dear Dr. Silveira,

We’re pleased to inform you that your manuscript has been judged scientifically suitable for publication and will be formally accepted for publication once it meets all outstanding technical requirements.

Kind regards,

Elingarami Sauli, PhD

Academic Editor

PLOS One
---

## [Editor Report · Acceptance letter]

PONE-D-24-40245R6

PLOS One

Dear Dr. Silveira,

I'm pleased to inform you that your manuscript has been deemed suitable for publication in PLOS One. Congratulations! Your manuscript is now being handed over to our production team.

Kind regards,

on behalf of

Dr. Elingarami Sauli

Academic Editor

PLOS One